# Antisense oligonucleotides to therapeutically target SARS-CoV-2 infection

Yuanyuan Qiao[1,2,3⊘], Jesse W. Wotring [4⊘], Charles J. Zhang[4], Xia Jiang[1], Lanbo Xiao[1], Andy Watt[5], Danielle Gattis[5], Eli Scandalis[5], Susan Freier[5], Yang Zheng[1], Carla D. Pretto[6], Stephanie J. Ellison[1], Eric E. Swayze[5], Shuling Guo[5], Jonathan Z. Sexton[4,6,7,8‡], Arul M. Chinnaiyan [1,2,3,9,10‡] *

1 Michigan Center for Translational Pathology, University of Michigan, Ann Arbor, MI, United States of America, 2 Department of Pathology, University of Michigan, Ann Arbor, MI, United States of America, 3 Rogel Cancer Center, University of Michigan, Ann Arbor, MI, United States of America, 4 Department of Medicinal Chemistry, College of Pharmacy, University of Michigan, Ann Arbor, MI, United States of America, 5 Ionis Pharmaceuticals, Carlsbad, CA, United States of America, 6 Department of Internal Medicine, University of Michigan, Ann Arbor, MI, United States of America, 7 Center for Drug Repurposing, University of Michigan, Ann Arbor, MI, United States of America, 8 Michigan Institute for Clinical and Health Research, University of Michigan, Ann Arbor, MI, United States of America, 9 Howard Hughes Medical Institute, University of Michigan, Ann Arbor, MI, United States of America, 10 Department of Urology, University of Michigan, Ann Arbor, MI, United States of America

⊘ These authors contributed equally to this work.
‡ JZS and AMC are co-senior authors on this work.
* arul@umich.edu

**Data Availability Statement:** All relevant data are within the manuscript and its Supporting Information files.

**Funding:** J.Z.S. is supported by the National Institute of Diabetes and Kidney Diseases

## Abstract

Although the COVID-19 pandemic began over three years ago, the virus responsible for the disease, SARS-CoV-2, continues to infect people across the globe. As such, there remains a critical need for development of novel therapeutics against SARS-CoV-2. One technology that has remained relatively unexplored in COVID-19 is the use of antisense oligonucleotides (ASOs)—short single-stranded nucleic acids that bind to target RNA transcripts to modulate their expression. In this study, ASOs targeted against the SARS-CoV-2 genome and host entry factors, *ACE2* and *TMPRSS2*, were designed and tested for their ability to inhibit cellular infection by SARS-CoV-2. Using our previously developed SARS-CoV-2 bioassay platform, we screened 180 total ASOs targeting various regions of the SARS-CoV-2 genome and validated several ASOs that potently blocked SARS-CoV-2 infection *in vitro*. Notably, select ASOs retained activity against both the WA1 and B.1.1.7 (commonly known as alpha) variants. Screening of *ACE2* and *TMPRSS2* ASOs showed that targeting of *ACE2* also potently prevented infection by the WA1 and B.1.1.7 SARS-CoV-2 viruses in the tested cell lines. Combined with the demonstrated success of ASOs in other disease indications, these results support further research into the development of ASOs targeting SARS-CoV-2 and host entry factors as potential COVID-19 therapeutics.

## Introduction

The COVID-19 pandemic has proven to be one of the greatest global public health challenges of modern times, already claiming the lives of millions of individuals worldwide since the first

(R01DK120623). J.W.W. is supported by an American Foundation for Pharmaceutical Education (AFPE) regional award. A.M.C. is a Howard Hughes Medical Institute Investigator, A. Alfred Taubman Scholar, and American Cancer Society Professor. The funders had no role in study design, data collection and analysis, decision to publish, or preparation of the manuscript.

**Competing interests:** I have read the journal's policy and the authors of this manuscript have the following competing interests: A.W., D.G., E.S., S.F., E.E.S., and S.G. are employees of Ionis Pharmaceuticals. The remaining authors have no competing interests. We did not receive any financial support from Ionis Pharmaceuticals; they just provided the antisense oligonucleotides for the study (ASOs).

reported cases in late 2019 [1]. Multiple coordinated efforts around the globe have since led to the rapid development and deployment of effective vaccines against SARS-CoV-2, the coronavirus responsible for COVID-19 [2–4]. Equally urgent research efforts have also focused on identifying effective therapeutics against SARS-CoV-2, leading to such agents as neutralizing monoclonal antibodies and antivirals [5–10]. Despite these advancements, several barriers still exist towards controlling the spread of SARS-CoV-2, including waning vaccine protection over time, lack of vaccine accessibility in certain geographic areas, inability of certain populations to receive vaccines, and the emergence of SARS-CoV-2 variants [11–14]. Thus, there remains a critical worldwide need for development of additional therapeutics to combat SARS-CoV-2.

Antisense oligonucleotides (ASOs) are a rapidly expanding class of therapeutics with clinical efficacy in multiple diseases [15, 16], including several that have received approval by the Food and Drug Administration (FDA): nusinersen for spinal muscular atrophy [17, 18]; eteplirsen [19], golodirsen [20], and viltolarsen [21] for Duchenne muscular dystrophy; inotersen for hereditary transthyretin-mediated amyloidosis [22]; volanosorsen for familial chylomicronemia syndrome [23]; and mipomersen for homozygous familial hypercholesterolemia [24]. ASOs are small, synthetic single-stranded nucleic acids that bind to their target RNA and modulate expression of the transcript through various mechanisms [15, 16]. Through classic Watson-Crick base pairing, ASOs can be designed to mediate RNase H-dependent cleavage and degradation of their target transcripts [22, 24]. Alternatively, binding of ASOs can lead to steric inhibition of translation and decreased expression of the target transcript [15, 16]. ASOs can also be engineered to bind specific sequences in transcripts and modulate alternative splicing [17–21]. Advances in medicinal chemistry in ASOs have improved their stability and biodistribution from first-generation phosphorothioate (PS) oligodeoxynucleotides to second-generation PS ASOs containing 2´ methoxyethyl (2´-MOE) and generation 2.5 PS ASOs containing 2´ constrained ethyl (2´cEt) modifications [15]. Generation 2.5 cEt ASOs have increased potency via higher affinity binding to target RNA [25, 26]. Since ASOs function by binding to specific RNA sequences, they can be designed to modulate the expression of any target, even those considered "undruggable" by conventional small-molecule methods that require a protein-binding surface [16].

SARS-CoV-2 cellular infection starts with a series of entry steps, beginning with binding of the viral spike (S) protein to the host receptor, angiotensin-converting enzyme 2 (ACE2) [27, 28]. Fusion of the viral and cellular membranes is aided by proteolytic cleavage of the S protein via transmembrane protease serine 2 (TMPRSS2) [29, 30]. In a previous study, we found that *ACE2* and *TMPRSS2* expressions were modulated by the androgen receptor (AR) in subsets of lung epithelial cells, and that therapies targeting AR or bromodomain and extraterminal domain (BET) proteins (regulators of AR activity) could decrease *ACE2* and *TMPRSS2* expressions [31]. Importantly, using a morphological cell profiling assay [32], we showed that AR and BET antagonists or degraders inhibited SARS-CoV-2 infection [31]. Here, we explored whether ASOs targeting *ACE2* and *TMPRSS2*, as well as ASOs targeting the positive-sense single-stranded RNA (ssRNA) SARS-CoV-2 genome, could prevent SARS-CoV-2 infection. We find that *ACE2* and SARS-CoV-2 ASOs are indeed effective at decreasing cellular infection by SARS-CoV-2, including infection by SARS-CoV-2 variants. Given their clinical success in other indications, these data suggest that ASOs targeting SARS-CoV-2 or host entry factors may be promising novel therapies for COVID-19.

## Results

ASOs targeting the positive-sense ssRNA SARS-CoV-2 genome were designed at Ionis using different chemistries, including 48 2'-O-methoxyethyl (2'MOE)-modified uniform antisense

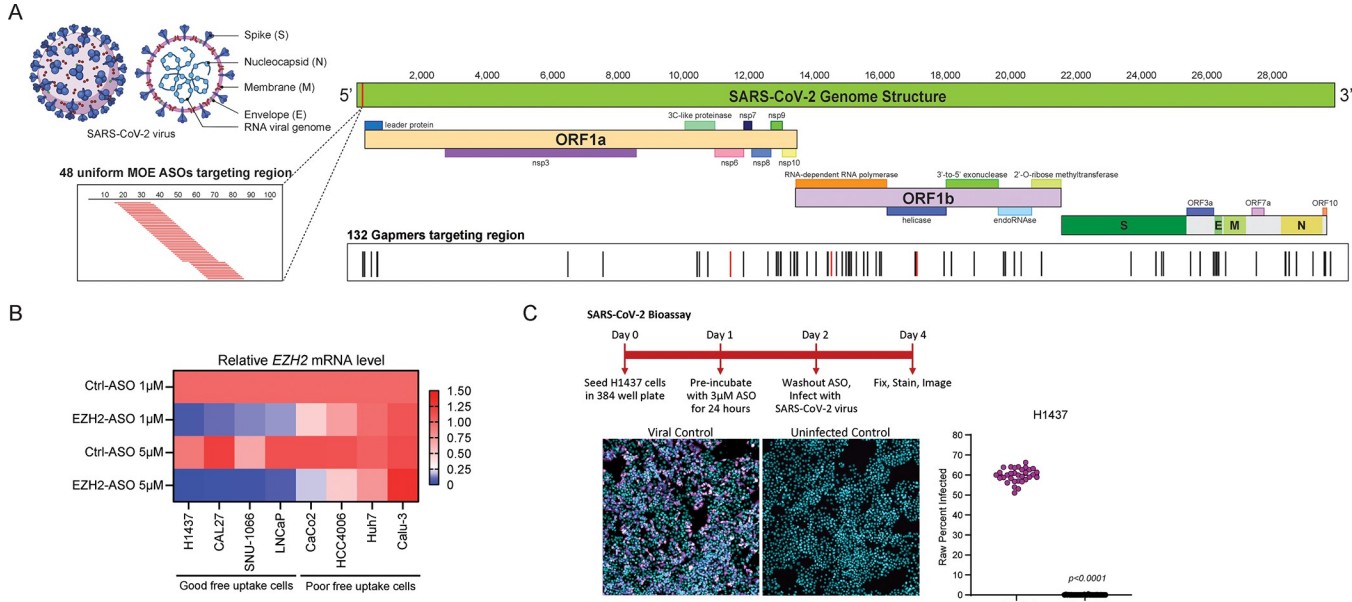

**Fig 1. Antisense oligonucleotides (ASOs) designed to target SARS-CoV-2 genome.** (A) Top left, SARS-CoV-2 viral structure. Right, SARS-CoV-2 genome structure and location of 48 2'-O-methoxyethyl (2'MOE)-modified uniform ASOs targeting the SARS-CoV-2 5' untranslated region (UTR) and 132 2´ constrained ethyl (cEt) Gapmer ASOs targeting additional regions. (B) Relative *EZH2* mRNA level in indicated cell lines after 1 µM or 5 µM of negative control (Ctrl) or *EZH2* ASO treatment showing the free uptake ability of various cell lines. (C) Biosafety Level 3 (BSL-3) SARS-CoV-2 bioassay in H1437 cells with washout step and raw infectivity of H1437 cells by SARS-CoV-2 WA1 strain.

oligonucleotide (ASO) sequences targeting the 5' untranslated region (UTR) for steric block-ing of translation initiation and 132 cEt Gapmer ASOs spanning the entire SARS-CoV-2 genome for RNase H1-mediated cleavage (Fig 1A, S1 Table). Prior to infection with the experi-mental SARS-CoV-2 ASOs, we first examined the ability of various cell lines to freely uptake ASOs using an ASO against the abundant, endogenous *EZH2* (Enhancer of Zeste 2 Polycomb Repressive Complex 2 Subunit) transcript compared with a non-targeting control, both at 1 and 5 µM. More than 75% downregulation of *EZH2* expression was achieved in certain cell lines, including H1437, CAL27, SNU-1066, and LNCaP, with either 1 or 5 µM *EZH2* ASO treatment. However, downregulation of *EZH2* was not observed in cell lines such as Caco2, HCC4006, Huh7, and Calu-3, even at 5 µM *EZH2* ASO treatment (Fig 1B). These results indi-cated that H1437, CAL27, SNU-1066, and LNCaP were competent ASO free uptake cell lines for the subsequent SARS-CoV-2 ASO screening experiments. However, amongst these free uptake favorable cell lines, we previously showed that only H1437 and LNCaP are permissive for SARS-CoV-2 infection [31]. Relevant to the study of SARS-CoV-2, H1437 is a lung lineage cell line, and our previously described SARS-CoV-2 bioassay platform [31, 32] showed that SARS-CoV-2 infection rates of 60% could be achieved in H1437 cells (Fig 1C); thus, H1437 cells were used for the subsequent SARS-CoV-2 ASO screening assays.

Initial screening assays were carried out by pretreating cells with 10 µM ASOs for 24 hours followed by infection with the WA1 strain of the SARS-CoV-2 virus for an additional 48 hours. The viral infectivity was calculated based on positive immunofluorescent staining of SARS-CoV-2 nuclear capsid count versus total nuclear count. Strong inhibition of SARS--CoV-2 infectivity in H1437 cells was observed with treatment of several ASOs targeting the 5'UTR region of the viral genome as well as Gapmer ASOs, with 50% of the screened ASOs showing more than 99% infectivity inhibition (S1A and S1B Fig). However, we noticed that non-viral targeting control ASOs also demonstrated strong viral infectivity inhibition when

cells were treated with 10 µM, but not 2 µM, ASOs in these conditions (S1C Fig), suggesting that high concentrations of ASOs might adversely affect SARS-CoV-2 infection of target cells and result in non-specific inhibition. To optimize the SARS-CoV-2 bioassay screening platform for ASOs, we thus examined whether an ASO washout step would increase the signal to noise ratio. Cells were preincubated with the most active ASOs from the initial screens (S1A and S1B Fig) for 24 hours, and then ASOs were washed out prior to the 48-hour SARS-CoV-2 infection. The results showed that the washout step successfully eliminated non-specific inhibition of SARS-CoV-2 infection, and the signal to noise ratio was largely improved (S1D Fig). Therefore, we repeated the screening of the 48 ASOs targeting the SARS-CoV-2 5'UTR and 132 Gapmer ASOs with the washout step in H1437 cells. The results indicated that Gapmer ASOs possessed stronger SARS-CoV-2 infectivity inhibition than the uniform MOE ASOs (S1E and S1F Fig).

The viral inhibition of SARS-CoV-2 by Gapmer ASOs having the greatest SARS-CoV-2 infectivity inhibition in the initial screening assay were subsequently validated using the SARS-CoV-2 bioassay with a washout step. The activity of the most active 26 SARS-CoV-2 Gapmers was confirmed against the WA1 virus strain in H1437 cells (Fig 2A). We further demonstrated that the three most active Gapmers (#13, 37, and 67) can inhibit infection of the WA1 strain in a dose-dependent manner in H1437 cells (Fig 2B and 2C). As further validation, we disrupted the gap of the ASOs by incorporating constrained ethyl modification in the gap so that they no longer support RNase H1 activity. These gap-ablated controls for the most active Gapmers were examined for WA1 strain infection in H1437 cells. The results showed that Gapmers #13, 37, 67, and 80 have strong viral inhibition ability against the WA1 strain, whereas the corresponding inactive forms of these Gapmers exhibited no such inhibition (Fig 2D). Gapmers #37, 67, and 80 remained effective in blocking infection of the SARS-CoV-2 B.1.1.7 variant (commonly known as the alpha variant, first identified in the United Kingdom [14]) in H1437 cells (Fig 2E). Additionally, the most active SARS-CoV-2 ASOs of 2'MOE chemistry targeting the 5'UTR also inhibited WA1 strain infection in a dose-dependent manner (Fig 2F). Together, these results demonstrate that SARS-CoV-2 Gapmers (#13, 37, 67, and 80) and uniform MOE ASOs (#1, 13) are strong inhibitors of cellular infection by multiple variants of SARS-CoV-2.

We next investigated whether ASOs targeting SARS-CoV-2 host entry factors *ACE2* and *TMPRSS2* could inhibit SARS-CoV-2 viral infection. As mentioned above, SARS-CoV-2 infection of host cells requires surface expression of ACE2 as a receptor for virus attachment, and then the viral S protein can be cleaved by TMPRSS2 to facilitate entry and release of the viral genome into host cells (Fig 3A, left). Several ASOs against *ACE2* and *TMPRSS2* were designed to target different regions of the *ACE2* and *TMPRSS2* genes, including intron and exon regions (Fig 3A, right). The knockdown efficacy of each ASO was examined by RT-qPCR with 1 and 3 µM ASO treatments in H1437 cells. The results showed that all nine *ACE2* ASOs decreased *ACE2* mRNA levels at both doses; additionally, nine *TMPRSS2* ASOs decreased *TMPRSS2* mRNA levels (Fig 3B). The infectivity of the WA1 strain in H1437 cells was examined with our SARS-CoV-2 bioassay in the presence of each *ACE2* or *TMPRSS2* ASO. Overall, the results demonstrated that *ACE2* ASOs exhibited greater viral inhibition than *TMPRSS2* ASOs against WA1 infection in H1437 cells (Fig 3C). *ACE2* ASOs were further confirmed to possess dose-dependent inhibition against the WA1 strain in H1437 cells (Fig 3D, top). Representative images indicated that the *ACE2* ASO #1 completely prevented WA1 infection of H1437 cells (Fig 3D, bottom). Furthermore, the gap-ablated inactive forms of *ACE2* ASOs showed that the viral inhibition effect was only observed with treatment of the active forms of *ACE2* ASOs against the WA1 strain (Fig 3E) and B.1.1.7 variant (Fig 3F).

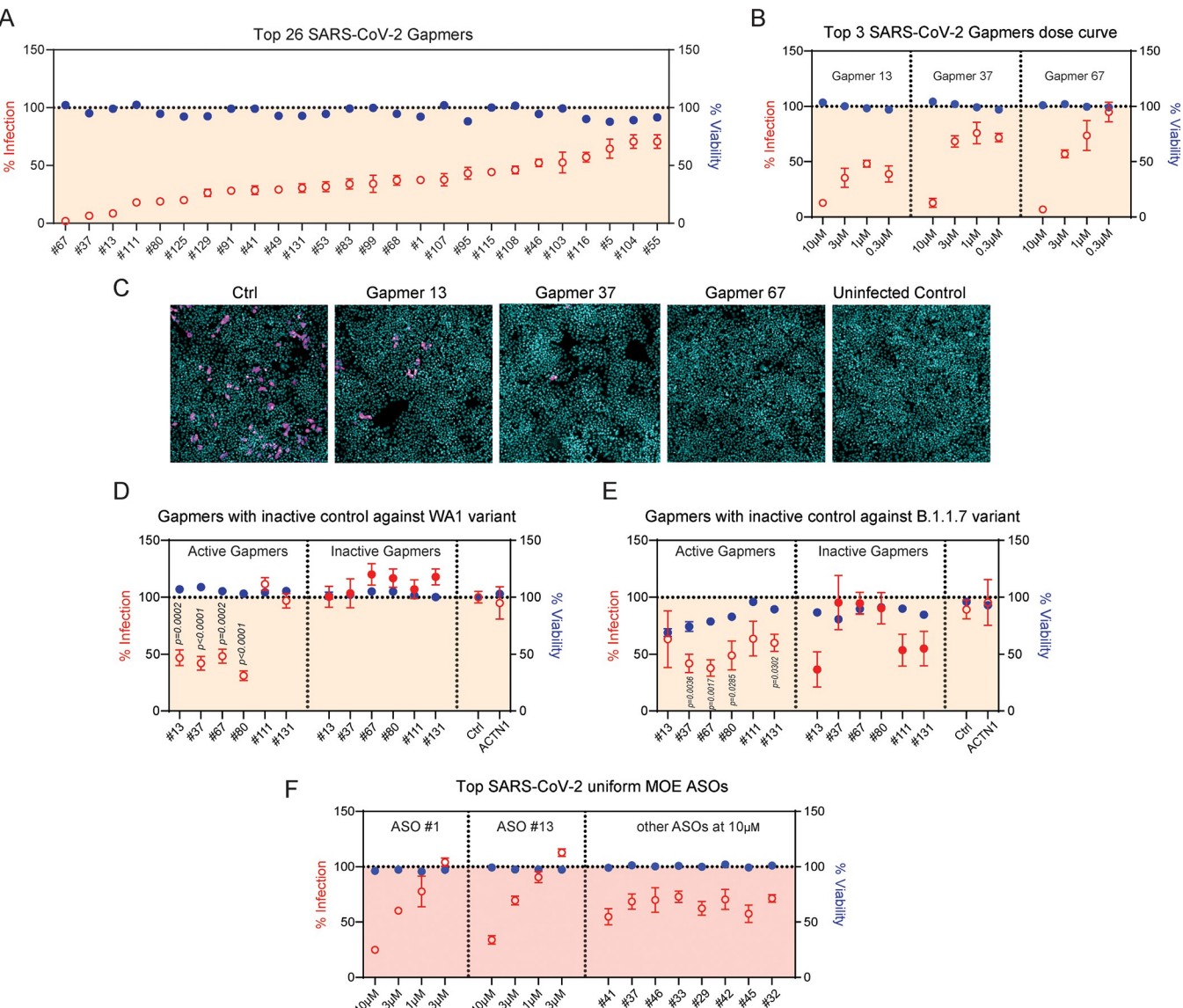

**Fig 2. Antiviral screen of SARS-CoV-2 ASOs in H1437 cells.** (A) SARS-CoV-2 bioassay showing infectivity and cell viability with the most active 26 SARS-CoV-2 Gapmers at 10 μM against WA1 strain in H1437 cells. (B) SARS-CoV-2 bioassay showing infectivity and cell viability with the most active 3 SARS-CoV-2 Gapmers at 10, 3, 1, and 0.3 μM treatments against WA1 strain in H1437 cells. (C) Representative images of H1437 cells after 10 μM indicated ASOs treatment. (D) SARS-CoV-2 bioassay showing infectivity and cell viability with the most active SARS-CoV-2 Gapmers and corresponding inactive controls at 3 μM against WA1 strain in H1437 cells. P values were calculated using two tailed unpaired *t* test by comparing to control (Ctrl). (E) SARS-CoV-2 bioassay showing infectivity and cell viability with the most active SARS-CoV-2 Gapmers and corresponding inactive controls at 3 μM against B.1.1.7 strain in H1437 cells. P values were calculated using two tailed unpaired *t* test by comparing to control (Ctrl). (F) SARS-CoV-2 bioassay showing infectivity and cell viability with the most active SARS-CoV-2 uniform MOE ASOs and corresponding inactive controls at 3 μM against WA1 strain in H1437 cells.

The inhibitory abilities of the most active *ACE2* ASOs and SARS-CoV-2 Gapmers were further examined using a physiologically relevant model of alveolar epithelial type 2 cells (iAEC2s) [32]. The majority of the *ACE2* ASOs remained efficacious in blocking SARS-CoV-2 infection by the B.1.1.7 variant when tested in iAEC2 cells, while select Gapmers targeting the SARS-CoV-2 genome also inhibited infection of iAEC2 cells (Fig 4A). Interestingly, ASOs targeting the androgen receptor (*AR*) also decreased B.1.1.7 variant infectivity in iAEC2 cells, suggesting infection of iAEC2 might be AR-regulated (Fig 4A). In summary, we have

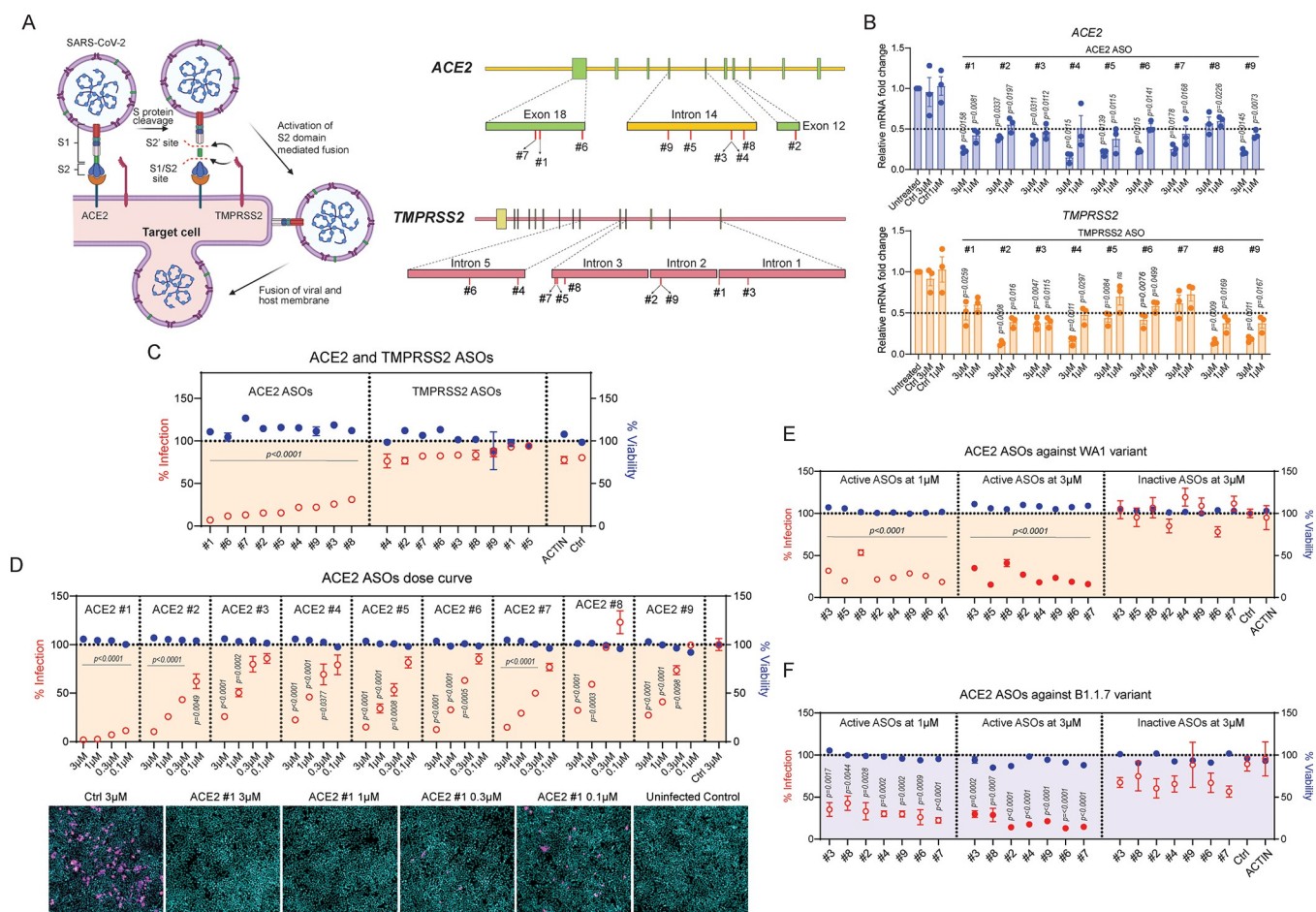

**Fig 3. Antiviral activity of ASOs targeting SARS-CoV-2 entry factors *ACE2* and *TMPRSS2*.** (A) Illustration of SARS-CoV-2 infection pathway of target cells (left). *ACE2* and *TMPRSS2* ASO targeting regions (right). (B) Relative *ACE2* and *TMPRSS2* mRNA expression levels after various ASO treatments at 3 μM or 1 μM in H1437 cells with washout step. P values were calculated using two tailed unpaired *t* test by comparing to control (Ctrl). (C) SARS-CoV-2 bioassay indicating infectivity and viability of H1437 cells after incubation with various *ACE2* and *TMPRSS2* ASOs against WA1 strain. P values were calculated using two tailed unpaired *t* test by comparing to control (Ctrl). (D) Dose response curve of SARS-CoV-2 bioassay of *ACE2* ASOs against WA1 strain in H1437 cells. Bottom, representative images of *ACE2* ASO #1 at varying concentrations. P values were calculated using two tailed unpaired *t* test by comparing to control (Ctrl). (E) Selected *ACE2* ASO active and inactive compounds against WA1 strain in H1437 cells. P values were calculated using two tailed unpaired *t* test by comparing to control (Ctrl). (F) Selected *ACE2* ASO active and inactive compounds against B.1.1.7 strain in H1437 cells. P values were calculated using two tailed unpaired *t* test by comparing to control (Ctrl).

demonstrated that ASOs targeting the SARS-CoV-2 host entry factor *ACE2* and SARS-CoV-2 genome can effectively inhibit cellular infection by various strains of SARS-CoV-2 *in vitro*, suggesting ASOs might be useful tools to prevent and reduce the burden of COVID-19 (Fig 4B).

## Discussion

Our previous work showed that infection by SARS-CoV-2 could be attenuated by treatment with AR or BET inhibitors, which decreased expression of the critical host entry factors *ACE2* and *TMPRSS2* [31]. In this study, we explored whether ASOs directly targeting host entry factors or the SARS-CoV-2 genome itself could prevent infection. Several ASOs designed to target *ACE2* and SARS-CoV-2 effectively inhibited SARS-CoV-2 infection, including infection by the B.1.1.7 (alpha) variant [14] (Fig 4B). ASOs targeting *TMPRSS2* were not as effective at

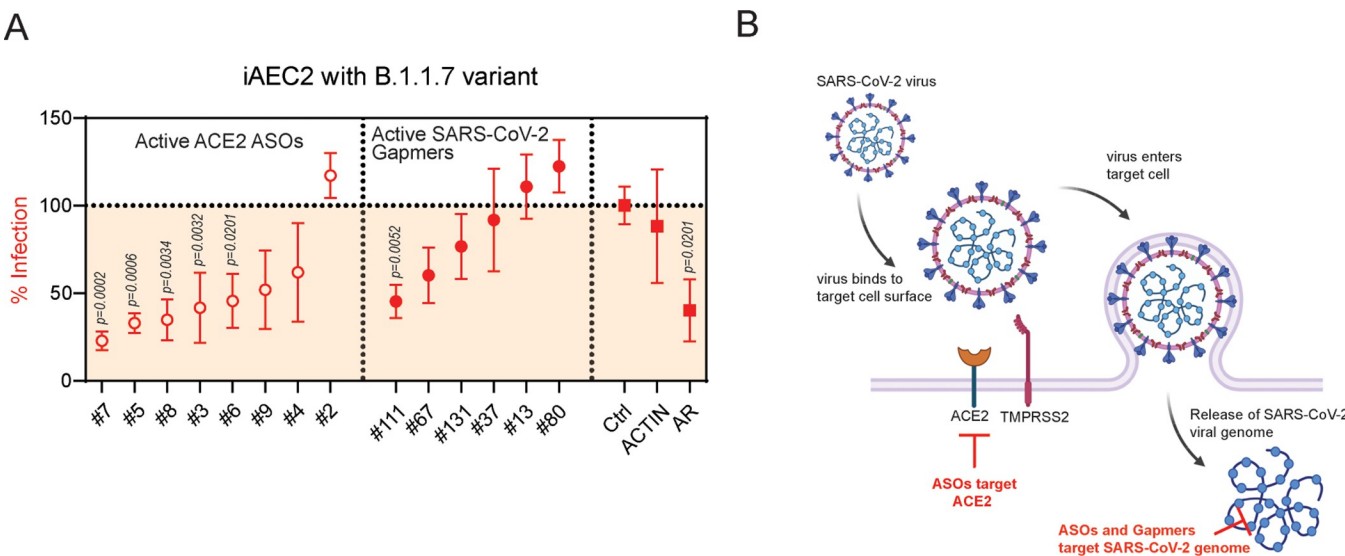

**Fig 4. Antiviral activities of *ACE2* and SARS-CoV-2 ASOs in iACE2 cells.** (A) SARS-CoV-2 bioassay of *ACE2* ASOs and SARS-CoV-2 Gapmers in iAEC2 cells against B.1.1.7 strain. P values were calculated using two tailed unpaired *t* test by comparing to control (Ctrl). (B) Schematic illustration of therapeutic potential of ASOs against SARS-CoV-2 infection.

preventing SARS-CoV-2 infection in H1437 cells, but this may be a cell line-dependent observation that should be examined in additional cell lines and lineages. Although the use of antisense drugs has been rapidly evolving in other disease areas [15, 16], the utility of ASOs for COVID-19 treatment has thus far been relatively unexplored. A recent study examining chimeric oligonucleotides, comprised of a 2'-*O*-methyl-modified ASO against the SARS-CoV-2 S transcript and a 5'-phosphorylated 2'-5' poly(A)$_4$ that guides RNase L into cleave the viral ssRNA, found that the chimeras inhibited infection in SARS-CoV-2 pseudovirus models [33]. An additional recent study suggests that ASOs targeting the frameshift stimulation element (FSE) of the SARS-CoV-2 genome may lead to inhibition of viral replication [34]. Another promising study with a locked nucleic acid (LNA) ASO targeting the SARS-CoV-2 5' leader sequence showed that viral replication could be inhibited *in vitro* as well as *in vivo* with daily intranasal ASO administration [35]. These data and the findings from our study suggest that ASO-based therapies should be evaluated further as possible treatments for COVID-19. ASOs can be effectively administered by several different routes, and relevant to COVID-19, this includes administration through aerosol routes to target pulmonary diseases [15, 36].

ASOs possess several advantages over other classes of therapeutics. ASOs may be designed based on the sequence of the specific target, however, as in our study, due to secondary and tertiary RNA structures that may preclude oligo binding, screening experiments are still often required to determine the most potent ASOs that can decrease target expression [37]. The sequence-based design of ASOs makes them particularly attractive therapeutics for directly targeting an RNA virus like SARS-CoV-2 as the sequence of the ASO can be easily modified to target any SARS-CoV-2 variants of concern that emerge over time [38]. Furthermore, because ASOs are sequence-specific, they are less likely to have off-target effects [16].

As briefly mentioned above, ASOs are also appealing agents since they can essentially be designed to target any transcript, including those conventionally considered "undruggable" by traditional methods. This includes a class of transcripts known as long non-coding RNAs (lncRNAs) that do not encode proteins but regulate critical cellular processes through various mechanisms and are associated with numerous diseases, such as cancer [39]. ASOs can also be

useful in targeting mRNAs encoding undruggable proteins that are difficult to inhibit at the protein level because they lack a binding surface for small-molecules [16]. As a further advantage, ASOs that are designed to decrease transcript expression (like *ACE2* ASOs in our study), either through RNase H-mediated cleavage or steric blocking of translation, prevent expression of the target protein altogether rather than only blocking protein activity like small-molecule inhibitors. The ability to design ASOs that modulate splicing patterns of pre-mRNAs is a further valuable characteristic. This has been shown to be clinically beneficial in different instances, including for the three FDA-approved ASOs for Duchenne muscular dystrophy, eteplirsen, golodirsen, and viltolarsen [19–21], which function by inducing exon skipping in the *DMD* transcript to produce a functional dystrophin protein that is otherwise lacking in the disease due to *DMD* gene mutations [40]. Altogether, these various aspects make ASOs desirable drugs to be further developed and clinically tested in COVID-19.

## Materials and methods

### Cell culture

H1437, CAL27, SNU-1066, HCC4006, Caco-2, Calu-3, Huh7, and LNCaP cells were obtained from ATCC and maintained under 5% $CO_2$ at 37˚C in medium according to ATCC's instructions. iAEC2 cells [iPSC (SPC2 iPSC line, clone SPC2-ST-B2, Boston University) derived alveolar epithelial type 2 cells] were maintained as previously described [32]. iAEC2 cells were subcultured every 2 weeks by dispase (2 mg/mL; Thermo Fisher Scientific; 17105–04) and 0.05% trypsin (Invitrogen; 25300054) digestion into single cells and replated in Matrigel (Corning; 356231). All cell lines tested negative for mycoplasma and were authenticated by genotyping.

### ASOs and treatment

All ASOs were designed, screened, characterized, and provided by Ionis Pharmaceuticals. The sequences of ASOs are listed in S1 Table. H1437 cells (3,000 per well) and iAEC2 cells (5,000 per well) were plated in 384-well plates (Perkin-Elmer, 6057300) overnight and then treated with ASOs for 24 hours prior to the SARS-CoV-2 bioassay. For washout experiments, ASOs were replaced with ASO-free medium after 24-hour ASO incubation right before the SARS-CoV-2 bioassay.

### SARS-CoV-2 high-content bioassay

SARS-CoV-2 isolates USA-WA1/2020 and USA/CA_CDC_5574_2020 (B.1.1.7) were obtained from BEI resources and propagated in VeroE6 cells (ATCC). Viral titers were determined by TCID50 using the Reed and Muench method. After ASO/Gapmer washout, H1437 or iAEC2 plates were transferred to a Biosafety Level 3 (BSL3) facility and infected with the indicated SARS-CoV-2 variant at an MOI = 1 for H1437 and MOI = 10 for iAEC2 in 10 μL complete cell culture medium for a total assay volume of 25+10 = 35 μL/well. Cells were then incubated for 48 hours post-infection at 37˚C and 5% $CO_2$. Assay plates were fixed, permeabilized, and labeled with anti-nucleocapsid SARS-CoV-2 primary antibody (Antibodies Online, Cat. #: ABIN6952432) as previously described [32]. Plates were surface decontaminated and transferred to a BSL2 lab where they were incubated overnight at 4˚C. Plates were then washed 2X with 25 μL/well PBS and stained with 25 μL/well of a fluorescent dye cocktail containing 1:1000 Hoechst 33342 to label nuclei and 1:1000 Alexa-647 secondary antibody to label virus (goat anti-mouse; Thermo Fisher; A21235) diluted in antibody buffer for 30 minutes in the dark at room temperature. Plates were then washed 2X with 25 μL/well PBS and stored in 25 μL/well PBS for fluorescence imaging.

## Fluorescence imaging and high-content analysis

Assay plates were imaged using a Thermo Fisher CX5 high-content microscope using LED excitation (386/23 nm, 650/13 nm) at 10X magnification. Five wells per condition and nine fields per well were imaged at a single Z-plane, which was determined by image-based auto-focus on the Hoechst 33342 channel. Exposure times were optimized using instrument software to maximize signal/noise ratios. Images were processed using CellProfiler 4.0 to determine raw percent infection and percent cell viability as previously described [41]. Infection and viability percentages were normalized to the average of the vehicle control on each plate.

## RNA isolation and quantitative real-time PCR

Total RNA was extracted from cells or tissue using the miRNeasy mini kit (Qiagen), and cDNA was synthesized from 1 μg total RNA using the High-Capacity cDNA Reverse Transcription Kit (Applied Biosystems). qPCR was performed using either fast SYBR green master mix or Taqman master mix (Applied Biosystems) on the ViiA7 Real-Time PCR System (Applied Biosystems). The target mRNA expression was quantified using the ΔΔCt method and normalized to *GAPDH* expression. The primer sequences used for the SYBR green qPCR are as follows: *EZH2*-Forward (GACCTCTGTCTTACTTGTGGAGC), *EZH2*- Reverse (CGTCAGATGGTGCCAGCAATAG); *ACE2*-Forward (TCCATTGGTCTTC TGTCACCCG), *ACE2*-Reverse (AGACCATCCACCTCCACTTCTC); *TMPRSS2*-Forward (CCTCTAACTGGTGTGATGGCGT), *TMPRSS2*-Reverse (TGCCAGGACTTCCTCTGAGATG); *GAPDH*-Forward (TGCACCACCAACTGCTTAGC), *GAPDH*-Reverse (GGCATGGACTGTG GTCATGAG).

## Supporting information

**S1 Fig. Washout step improves signal to noise ratio of ASOs in the SARS-CoV-2 bioassay.** (A) SARS-CoV-2 bioassay screening for 48 uniform MOE ASOs targeting SARS-CoV-2 genome against WA1 strain in H1437 cells without washout step. (B) SARS-CoV-2 bioassay screening for 132 Gapmers targeting SARS-CoV-2 genome against WA1 strain in H1437 cells without washout step. (C) SARS-CoV-2 bioassay for control (Ctrl) ASO and *ACTN1* ASO without washout step against WA1 strain in H1437 cells showing non-specific activity at 10 μM. (D) Comparison of selected SARS-CoV-2 ASOs with or without washout step against WA1 strain in H1437 cells. (E) SARS-CoV-2 bioassay screening for 48 uniform MOE ASOs targeting SARS-CoV-2 genome with washout step against WA1 strain in H1437 cells. (F) SARS-CoV-2 bioassay screening for 132 Gapmers targeting SARS-CoV-2 genome with washout step against WA1 strain in H1437 cells.
(TIF)

**S1 Table. ASO sequences used in this study.** Sequences for ASOs targeting the SARS-CoV-2 genome or host genes are presented in the Excel file.
(XLSX)

## Acknowledgments

The following reagents were deposited by Centers for Disease Control and Prevention and obtained through BEI resources, NIAID, NIH: SARS-Related Coronavirus 2, Isolate USA-WA1/2020 (NR-52281) and USA/CA_CDC_5574/2020 (NR-54011).

## Author Contributions

**Conceptualization:** Yuanyuan Qiao, Jesse W. Wotring, Jonathan Z. Sexton, Arul M. Chinnaiyan.

**Data curation:** Yuanyuan Qiao, Jesse W. Wotring.

**Formal analysis:** Yuanyuan Qiao, Jesse W. Wotring, Charles J. Zhang, Xia Jiang, Lanbo Xiao, Andy Watt, Danielle Gattis, Eli Scandalis, Susan Freier, Carla D. Pretto, Eric E. Swayze, Shuling Guo, Jonathan Z. Sexton, Arul M. Chinnaiyan.

**Investigation:** Yuanyuan Qiao, Jesse W. Wotring, Charles J. Zhang, Xia Jiang, Lanbo Xiao, Andy Watt, Danielle Gattis, Eli Scandalis, Susan Freier, Carla D. Pretto, Eric E. Swayze, Shuling Guo.

**Methodology:** Andy Watt, Danielle Gattis, Eli Scandalis, Susan Freier, Eric E. Swayze, Shuling Guo.

**Supervision:** Jonathan Z. Sexton, Arul M. Chinnaiyan.

**Visualization:** Yang Zheng.

**Writing – original draft:** Yuanyuan Qiao, Jesse W. Wotring, Stephanie J. Ellison, Jonathan Z. Sexton, Arul M. Chinnaiyan.

**Writing – review & editing:** Yuanyuan Qiao, Jesse W. Wotring, Stephanie J. Ellison, Jonathan Z. Sexton, Arul M. Chinnaiyan.

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
