## [Decision Letter · Decision Letter 0]

1 Dec 2022

PONE-D-22-30002Antisense oligonucleotides to therapeutically target SARS-CoV-2 infectionPLOS ONE

Dear Dr. Chinnaiyan,

Thank you for submitting your manuscript to PLOS ONE. After careful consideration, we feel that it has merit but does not fully meet PLOS ONE’s publication criteria as it currently stands. Therefore, we invite you to submit a revised version of the manuscript that addresses the points raised during the review process.

Your manuscript has been reviewed by two external reviewers and they both suggested major revision. New experiments may be necessary to satisfactorily address all the reviewer's concerns. In addition, one of the reviewers think additional statistical analysis is necessary, while the other suggested that the manuscript needs English language corrections.

We look forward to receiving your revised manuscript.

Kind regards,

Arunava Roy, Ph.D.

Academic Editor

PLOS ONE

Journal Requirements:

"I have read the journal's policy and the authors of this manuscript have the following competing interests: A.W., E.S., S.G., and S.F. are employees of Ionis Pharmaceuticals. The remaining authors have no competing interests."

We note that you received funding from a commercial source: Ionis Pharmaceuticals

"The following reagents were deposited by Centers for Disease Control and Prevention and

obtained through BEI resources, NIAID, NIH: SARS-Related Coronavirus 2, Isolate USAWA1/2020 (NR-52281) and USA/CA_CDC_5574/2020 (NR-54011). J.Z.S. is supported by the

National Institute of Diabetes and Kidney Diseases (R01DK120623). J.W.W. is supported by an

American Foundation for Pharmaceutical Education (AFPE) regional award. A.M.C. is a Howard

Hughes Medical Institute Investigator, A. Alfred Taubman Scholar, and American Cancer Society

Professor."

"The following reagents were deposited by Centers for Disease Control and Prevention and obtained through BEI resources, NIAID, NIH: SARS-Related Coronavirus 2, Isolate USA-WA1/2020 (NR-52281) and USA/CA_CDC_5574/2020 (NR-54011). J.Z.S. is supported by the National Institute of Diabetes and Kidney Diseases (R01DK120623). J.W.W. is supported by an American Foundation for Pharmaceutical Education (AFPE) regional award. A.M.C. is a Howard Hughes Medical Institute Investigator, A. Alfred Taubman Scholar, and American Cancer Society Professor."

Reviewers' comments:

Reviewer's Responses to Questions

**Comments to the Author**

1. Is the manuscript technically sound, and do the data support the conclusions?

Reviewer #1: Yes

Reviewer #2: Yes

2. Has the statistical analysis been performed appropriately and rigorously? 

Reviewer #1: Yes

Reviewer #2: No

3. Have the authors made all data underlying the findings in their manuscript fully available?

Reviewer #1: Yes

Reviewer #2: Yes

4. Is the manuscript presented in an intelligible fashion and written in standard English?

Reviewer #1: No

Reviewer #2: Yes

5. Review Comments to the Author

Reviewer #1: The authors have shown that ASOs -Gapmers and MOE modified ASOs are strong inhibitors of SARS-COV2 infection. The authors need to perform further experiments to validate their findings.

1. Did the authors check for toxicity of these ASOs?

2. What was the duration and MOI for virus infection?

3. Do authors have in vivo proof of concept for their ASOs?

4. Abstract is very incoherent. Its been a few years not several years of pandemic.

5. Authors should also check for virus titers after treating in with ASOs.

Reviewer #2: The manuscript titled "Antisense oligonucleotides to therapeutically target SARS-CoV-2 infection" is well written in English. The authors screen ACE2 and TMPRSS2 ASOs to inhibit the infection of SARS-CoV-2 variants. The study demonstrates the usefulness of the approach and it can be used in other disease models as it is a very specific response unlike certain drugs.

The authors have not performed any statistical analysis which is visible on the graphs for reduction of viral infection using the ASOs, which is a major flaw to understand of the data is significant.

6. PLOS authors have the option to publish the peer review history of their article (what does this mean?). If published, this will include your full peer review and any attached files.

Reviewer #1: No

Reviewer #2: **Yes: **Dr. SNEHA SINGH

---

## [Author Response · Author response to Decision Letter 0]

9 Jan 2023

Please see cover letter and response to reviewer files included with the package

---

## [Editor Report · Decision Letter 1]

13 Jan 2023

PONE-D-22-30002R1Antisense oligonucleotides to therapeutically target SARS-CoV-2 infectionPLOS ONE

Dear Dr. Chinnaiyan,

Thank you for submitting your manuscript to PLOS ONE. After careful consideration, we feel that it has merit but does not fully meet PLOS ONE’s publication criteria as it currently stands. Therefore, we invite you to submit a revised version of the manuscript that addresses the points raised during the review process.

1) How many biological replicates were performed for each experimental condition in all the figures? I did not find this information in the manuscript. This information needs to be mentioned either in the figure legend or in the M&M.

You mentioned, "Nine fields per well were imaged at a single Z-plane, which was determined by image-based autofocus on the Hoechst 33342 channel". These are technical replicates. Please mention the number of wells imaged for each experiment.

2) What plates were used for the SARS-CoV-2 bioassay? The M&M mentions, "Columns 23 and 24 were always left uninfected for mock controls". However, without a description of the plate used, this information is unnecessary.

3) According to your M&M, the virus was allowed to infect the cells for 48 hours without changing the medium. For SARS-CoV-2 research, it's common practice to wash and change the medium after 2 hours of infection. Thereafter, the cells can be incubated for 48 hours as required. Please confirm what was done.

We look forward to receiving your revised manuscript.

Kind regards,

Arunava Roy, Ph.D.

Academic Editor

PLOS ONE
---

## [Author Response · Author response to Decision Letter 1]

17 Jan 2023

Please see the cover letter document for responses to editor comments and the response to reviewers document for answers to reviewers' questions.

---

## [Editor Report · Decision Letter 2]

20 Jan 2023

Antisense oligonucleotides to therapeutically target SARS-CoV-2 infection

PONE-D-22-30002R2

Dear Dr. Chinnaiyan,

We’re pleased to inform you that your manuscript has been judged scientifically suitable for publication and will be formally accepted for publication once it meets all outstanding technical requirements.

Kind regards,

Arunava Roy, Ph.D.

Academic Editor

PLOS ONE

---

## [Editor Report · Acceptance letter]

26 Jan 2023

PONE-D-22-30002R2 

Antisense oligonucleotides to therapeutically target SARS-CoV-2 infection 

Dear Dr. Chinnaiyan:

I'm pleased to inform you that your manuscript has been deemed suitable for publication in PLOS ONE. Congratulations! Your manuscript is now with our production department. 

Kind regards, 

on behalf of

Dr. Arunava Roy 

Academic Editor

PLOS ONE